# Epidemiology of Methicillin-Resistant *Staphylococcus aureus* in Slovakia, 2020 – Emergence of an Epidemic USA300 Clone in Community and Hospitals

Jan Tkadlec,[a] Anh Vu Le,[b] Marie Brajerova,[a] Anna Soltesova,[c] Jozef Marcisin,[d] Pavel Drevinek,[a] Marcela Krutova[a]

[a]Department of Medical Microbiology, Charles University, 2nd Faculty of Medicine and Motol University Hospital, Prague, Czech Republic
[b]Department of Computer Science, Czech Technical University, Faculty of Electrical Engineering, Prague, Czech Republic
[c]Department of Clinical Biochemistry, Haematology and Microbiology, Unilabs Slovensko, s.r.o., Roznava, Slovakia
[d]Department of Clinical Microbiology, Unilabs Slovensko, s.r.o., Stropkov, Slovakia

**ABSTRACT**  Methicillin-resistant *Staphylococcus aureus* (MRSA) is a leading cause of health care-associated infections. Additionally, over the decades, the spread of community-associated (CA-MRSA) clones has become a serious problem. The aim of this study was to gain data on the current epidemiology of MRSA in Slovakia. Between January 2020 and March 2020, single-patient MRSA isolates (invasive and/or colonizing) were collected in Slovakia from hospitalized inpatients (16 hospitals) or outpatients (77 cities). Isolates were characterized via antimicrobial susceptibility testing, *spa* typing, SCC*mec* typing, the detection of *mecA/mecC*, genes coding for Panton-Valentine leukocidin (PVL), and the *arcA* gene (part of the arginine catabolic mobile element [ACME]). Out of 412 isolates, 167 and 245 originated from hospitalized patients and outpatients, respectively. Inpatients were most likely older ($P < 0.001$) and carried a strain exhibiting multiple resistance ($P = 0.015$). Isolates were frequently resistant to erythromycin ($n = 320$), clindamycin ($n = 268$), and ciprofloxacin/norfloxacin ($n = 261$). 55 isolates were resistant to oxacillin/cefoxitin only. By clonal structure, CC5-MRSA-II ($n = 106$; *spa* types t003, t014), CC22-MRSA-IV ($n = 75$; t032), and CC8-MRSA-IV ($n = 65$; t008) were the most frequent. We identified PVL in 72 isolates (17.48%; 17/412), with the majority belonging to CC8-MRSA-IV ($n = 55$; *arcA*+; t008, t622; the USA300 CA-MRSA clone) and CC5-MRSA-IV ($n = 13$; t311, t323). To the best of our knowledge, this is the first study on the epidemiology of MRSA in Slovakia. The presence of the epidemic HA-MRSA clones CC5-MRSA-II and CC22-MRSA-IV was found, as was, importantly, the emergence of the global epidemic USA300 CA-MRSA clone. The extensive spread of USA300 among inpatients and outpatients across the Slovakian regions warrants further investigation.

**IMPORTANCE**  The epidemiology of MRSA is characterized by the rise and fall of epidemic clones. Understanding the spread, as well as the evolution of successful MRSA clones, depends on the knowledge of global MRSA epidemiology. However, basic knowledge about MRSA epidemiology is still fragmented or completely missing in some parts of the world. This is the first study of MRSA epidemiology in Slovakia to identify the presence of the epidemic HA-MRSA clones CC5-MRSA-II and CC22-MRSA-IV and, importantly and unexpectedly, the emergence of the global epidemic USA300 CA-MRSA clone in the Slovakian community and hospitals. So far, USA300 has failed to spread in Europe, and this study documents an extensive spread of this epidemic clone in a European country for the first time.

**KEYWORDS**  MRSA, PVL, USA300, Slovakia, CA-MRSA, *spa* typing

Address correspondence to Jan Tkadlec, jan.tkadlec@lfmotol.cuni.cz.

The authors declare a conflict of interest. Pavel Drevinek received research funding from the Ministry of Health, Czech Republic, consultation fees and honoraria for lectures from Vertex Pharmaceuticals and Viatris, support for attending meetings from I.T.A. Interact s.r.o., Chiesi CZ. Pavel Drevinek is also president of the Czech Society for Medical Microbiology and the board member of the European Cystic Fibrosis Society. The sponsoring institutions had no role in the study design, data collection, analysis, and interpretation of data as well as in the writing of the manuscript. The remaining authors declare that they have no conflicts of interest.

Methicillin-resistant *Staphylococcus aureus* (MRSA) is a notorious nosocomial pathogen that causes a globally significant burden on the health care system (1, 2). In the late 1990s, distinct strains associated with community spreading emerged, changing the traditional

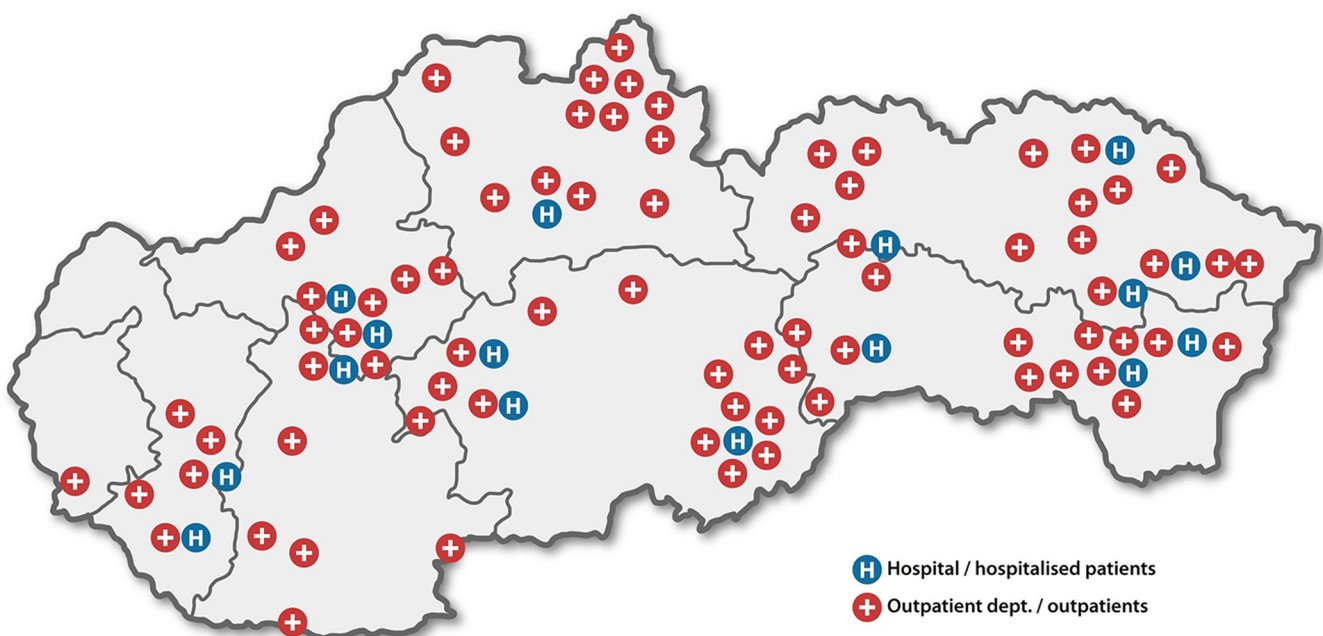

**FIG 1** Distribution of participating hospitals and outpatient departments across administrative regions of Slovakia.

view of MRSA as a nosocomial pathogen (3, 4). Community-associated MRSA (CA-MRSA) strains have become common in the last two decades. CA-MRSA strains are frequently associated with increased virulence and genetic factors, such as Panton-Valentine leukocidin (PVL). Some CA-MRSA lineages also possess arginine catabolic mobile elements (ACME) that are responsible for increased survival on human skin. The distinctive feature of CA-MRSA is its capacity to cause severe infection in young and healthy people, compared to health-care-associated MRSA (HA-MRSA), which affects patients of advanced age as well as those with comorbidities (4).

The population structure of MRSA is characterized by the presence of epidemic clones. The selective pressure of antibiotics, along with the subsequent acquisition of resistance, is the main contributing factor to the expansion of epidemic clones, such as the global expansion of ST5-MRSA-II (5) or ST8-MRSA-IV (USA300 [CC8]) (6).

Slovakia has been a participating country in the European monitoring of the resistance prevalence among bacterial isolates that cause invasive infections European Antimicrobial Resistance Surveillance Network (EARS-Net) since 2011. According to EARS-NET data, the frequency of MRSA in Slovakia is relatively high, ranging between 25% to 30% of invasive staphylococcal isolates between 2016 and 2020 (7). However, little is known about the population structure of MRSA clones circulating in the country. Only one isolate from Slovakia was investigated in the study by Rolo et al., which focused on community-associated *S. aureus* across Europe (8). Following our previous study of MRSA epidemiology in the Czech Republic (9), we aimed to collect and characterize the representative number of MRSA isolates circulating in the community and hospitals in Slovakia. We want to use the acquired data to fill the gap in Slovak MRSA epidemiology and compare it with the situation in the Czech Republic as well as other neighboring countries.

## RESULTS

**Patient data.** In total, 412 single-patient MRSA isolates were collected in Slovakia, and these came from 167 inpatients hospitalized in 16 hospitals and 245 outpatients attending health care facilities in 77 cities (Fig. 1).

The isolates originated from the following types of samples: nose swab ($n = 187$), wound swab ($n = 114$), throat swab ($n = 54$), genitourinary tract samples ($n = 14$), ear swab ($n = 10$), lower airway samples ($n = 10$), blood cultures ($n = 7$), catheters ($n = 4$), eye swab ($n = 4$), and others ($n = 8$). In 165 (40.05%; 165/412) and 243 (58.98%; 243/412) of the cases, the isolates

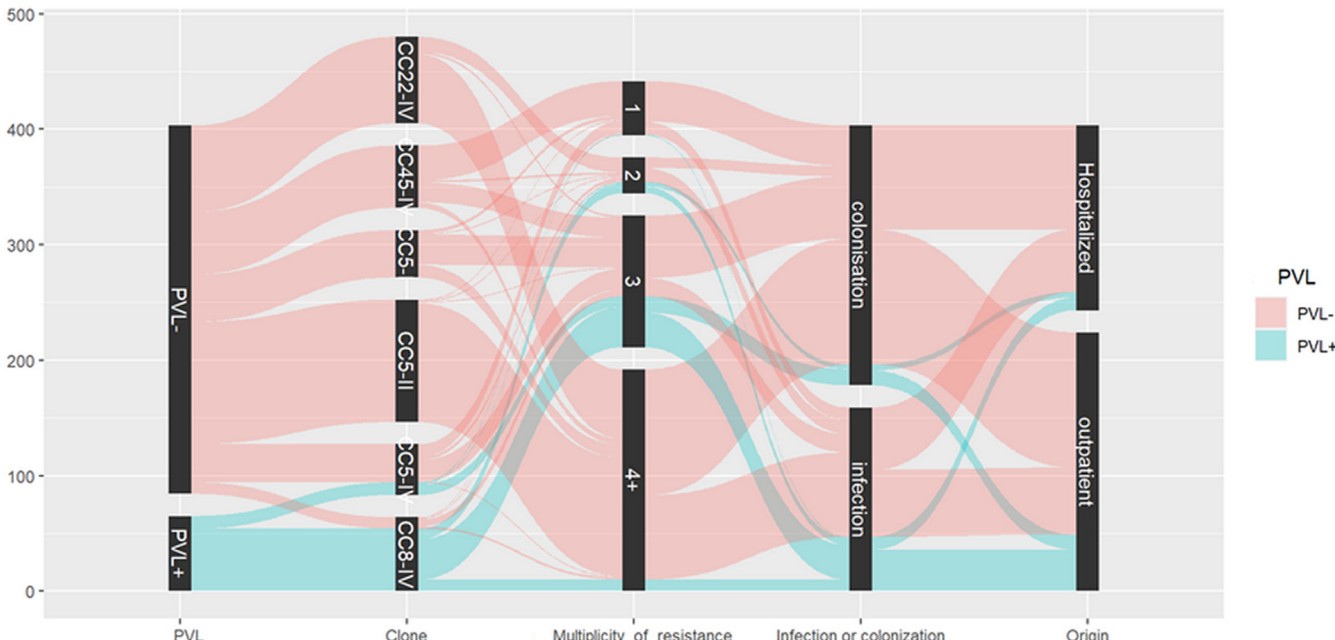

**FIG 2** Parallel coordinates describing the association of the presence of PVL and MRSA clones, the levels of resistance, and the origin of infection or colonization. The multiplicity of resistance was based on the categorization of antimicrobial classes used for the definition of multidrug-resistant, extensively drug-resistant and pandrug-resistant bacteria (29). Outpatient includes patients attending the emergency departments of the participating hospitals.

were considered to be the cause of infection and asymptomatic carriage, respectively. For four isolates, assignment to an invasive or colonizing group was not possible.

The median age of patients was 45 years (range of 1 day to 98 years), and 199 (48.30%; 199/412) of the patients were men. For men and women, there was no statistically significant difference regarding the age, region, infection/colonization status, origin of the isolate, clonal lineage, or PVL positivity of the isolate. The inpatients (median age of 67.49 years) and patients with MRSA infections (59.36 years) were significantly older than were the outpatients (27.47 years; $P < 0.001$) and the patients with asymptomatic colonization (32.64 years; $P = 0.002$). For the associations of various isolate characteristics, see Fig. 2.

**Antimicrobial susceptibility.** Besides oxacillin and cefoxitin resistance, which were detected in all cases, the MRSA isolates were frequently resistant to erythromycin (77.67%; 320/412), clindamycin (65.05%; 268/412), and norfloxacin/ciprofloxacin (63.35%; 261/412). Resistance to clindamycin was inducible in 44.40% (119/268) of the clindamycin-resistant isolates. Resistance to fusidic acid, tetracycline, and gentamicin was presented in 18 (4.37%; 18/412), 15 (3.64%; 15/412), and 12 (2.91%; 12/412) isolates, respectively. Resistance to rifampin ($n = 4$) and mupirocin ($n = 3$) were presented in fewer than 1% of isolates. None of the isolates were resistant to sulfamethoxazole/trimethoprim, linezolid, and ceftaroline. The gradient diffusion method confirmed resistance to tigecycline, as observed via disc diffusion with a MIC of 1 mg/L. On average, isolates were resistant to 3 different classes of antimicrobials, but 55 (13.32%; 55/412) isolates were resistant to oxacillin and cefoxitin only (for the resistance profiles, see Table 1). Isolates resistant to four or more classes (median age of 63.16 years) of antimicrobials originated from significantly older patients ($P < 0.001$) and were associated with inpatients ($P = 0.015$).

**Clonal structure.** Among the 412 isolates, 66 *spa* types were identified. 28 *spa* types were present among inpatients, and 53 *spa* types were present among outpatients. Isolates carried SCC*mec* cassette type I ($n = 41$), type II ($n = 106$), type IV ($n = 248$), and type V ($n = 9$). For 404 isolates, the MLST clonal complexes (CC) derived from the *spa* types and the SCC*mec* types were available. A total of 14 MRSA clonal lineages were identified. Six major clonal lineages, each representing more than 40 isolates, accounted for 94.17% (388/412) of the isolates (Table 2).

The CC5-MRSA-II was the most common MRSA lineage (25.73%; 106/412). It was present in all regions and dominating in both inpatients (32.34%; 54/167) and outpatients (21.22%;

**TABLE 1** Antimicrobial resistance profiles of MRSA isolates, according to patient care setting (Slovakia, January to March of 2020)[a]

| Resistance profile[b] | Inpatients (n = 167) | | Outpatients (n = 245) | | Total (n = 412) |
|---|---|---|---|---|---|
| | No. (%) | spa types (no) | No. (%) | spa types (no) | No. (%) |
| ERY, CLI, F | 87 (52.10) | t002(1); t003(15); t010(1); t014(30); t022(1); t025(1); t032(33); t535(1); t586(1); t4232(1); t10016(1); t19657(1) | 83 (33.88) | t002(1); t003(20); **t008**(3); t010(6); t014(23); t020(1); t022(1); t026(2); t032(15); t045(2); t264(1); t330(2); t910(1); t1282(2); tt2892 (1); t19658(1); t19660(1) | 170 (41.26) |
| ERY, CLI | 20 (11.98) | t010(2); t026(3); t179(5); **t311**(7); t855(1); t1683(1); **t3235**(1) | 55 (22.45) | **t008**(1); t010(19); t015(4); t026(2); t065(1); t179(1); **t311**(10); t330(6); t706(1); t1215(1); t1303(1); t1683(1); t2065(1); **t3235**(2); t4545(1); t9710(1); t11862(1); t19662(1) | 75 (18.20) |
| All susceptible | 18 (10.78) | t010(1); t026(5); t095(1); t179(1); **t311**(3); t586(1); t671(1); t779(1); t12321(3); t19661(1) | 37 (15.10) | t002(1); t010(2); t015(2); t024(1); t026(6); t065(1); t311(5); t548(1); t671(12); t880(1); t1265(1); t1345(1); t4264(1); t7603(1); t17768(1) | 55 (13.35) |
| ERY, F | 12 (7.19) | t003(1); **t008**(11) | 29 (11.84) | **t008**(26); **t622**(2); t19659(1) | 41 (9.95) |
| F | 12 (7.19) | t003(1); **t008**(4); t032(5); t1302(1); t6589(1) | 12 (4.90) | **t008**(3); t010(1); t015(1); t032(4); t179(1); t6589(1); t19753(1) | 24 (5.83) |
| ERY, CLI, F, GEN | 2 (1.20) | t003(2) | 3 (1.22) | t003(1); **t008**(2) | 5 (1.21) |
| ERY, CLI, TET | 1 (0.60) | t034 | 4 (1.63) | t034(1); t127(1); **t2393**(1); t17565(1) | 5 (1.21) |
| ERY | 1 (0.60) | t026 | 4 (1.63) | **t008**(3); t311(1) | 5 (1.21) |
| ERY, F, FUS | 0 (0.00) | | 4 (1.63) | **t008**(3); t316(1) | 4 (0.97) |
| F, FUS | 3 (1.80) | t026(2); t032(1) | 1 (0.41) | t032(1) | 4 (0.97) |
| ERY, CLI, F, GEN, TET | 1 (0.60) | t032 | 2 (0.81) | **t008**(1); t014(1) | 3 (0.73) |
| Other combinations[c] | 10 (5.99) | | 11 (4.49) | | 21 (5.10) |

[a]spa types corresponding to PVL-positive isolates are in bold and underlined.
[b]All isolates were resistant to oxacillin and cefoxitin. ERY, erythromycin; CLI, clindamycin; F, fluoroquinolones (norfloxacin and/or ciprofloxacin); GEN, gentamicin; TET, tetracycline; FUS, fusidic acid.
[c]For other combinations see the Supplemental Material. spa types corresponding to PVL-positive isolates are in bold and underlined.

52/245), with no statistical difference between invasive and colonizing strains ($P = 1.000$). The majority of isolates belonged to spa types t014 ($n = 55$) and t003 ($n = 42$), which are representatives of the Rhine-Hesse MRSA clone (ST225-MRSA-II). Patients infected or colonized by CC5-MRSA-II (median age of 70.8 years) were significantly older ($P < 0.001$), compared to patients with other lineages, with the exception of CC22-MRSA-IV ($P = 1.000$) (for a boxplot comparing the age distribution of the patients among MRSA clones, see the Supplemental Material). The CC5-MRSA-II isolates were resistant to four or more categories of antibiotics ($P < 0.001$). The typical resistance profile (97.17%; 103/106) was the combination of resistance to antistaphylococcal beta-lactams, macrolides, fluoroquinolones, and lincosamides.

The second most common lineage was CC22-MRSA-IV (18.20%; 75/412), which is known also as an EMRSA-15 or Barnim clone. All isolates carried SCCmec subtype IVh, and the most frequent spa type was t032 ($n = 65$), whereas other spa types were rare. CC22-MRSA-IV was associated more with inpatients (47 of 75 isolates; $P < 0.001$) and older age (median age of 67.28 years, $P < 0.001$), compared to other lineages, with the exception of CC5-MRSA-II. All of the isolates were resistant to fluoroquinolones. In 60 (80.00%; 60/75) isolates, a combined resistance to macrolides and lincosamides was detected.

The third most frequently detected lineage was CC8-MRSA-IV (15.78%; 65/412). SCCmec type IVa and IVc were carried by 60 and 2 isolates, respectively, and the subtypes of 3 isolates were not determined. The arcA gene, which is a part of the ACME mobile genetic element, was detected in 60 isolates, and PVL toxin genes were carried by 55 isolates. CC8-MRSA-IV isolates were frequently resistant to norfloxacin/ciprofloxacin (90.77%; 59/65) and erythromycin (87.69%; 57/65) but rarely to clindamycin (12.31%; 8/65). Analyzing 64 isolates with the complete data, the median age of patients with CC8-MRSA-IV was 31.47 years, making them significantly younger, compared to the CC5-MRSA-II and CC22-MRSA-IV patients (both $P < 0.001$). Among the isolates originating from emergency department patients ($n = 8$), half of them belonged to CC8-MRSA-IV, and all of them were isolated from wound swabs. In general, the CC8-MRSA-IV isolates were associated with patients with infection ($n = 43$;

**TABLE 2** Clonal structure of MRSA isolates (Slovakia, January to March 2020)[a]

| Clone | No. (%) | Inpatient (%) | Outpatient (%) | PVL (%) | AR profile (%) | Most frequent spa types | | | |
|---|---|---|---|---|---|---|---|---|---|
| | | | | | | 1st (no.) | 2nd (no.) | 3rd (no.) | Others (no.) |
| CC5-MRSA-II | 106 (25.73) | 54 (32.34) | 52 (21.22) | 0 (0.00) | ERY, CLI, F (90.57) | t014(55) | t003(42) | t002(2); t586(2) | t264(1); t535(1); t1282(1); t19658(1); t19660(1); |
| CC22-MRSA-IV | 75 (18.20) | 47 (28.14) | 28 (11.43) | 0 (0.00) | ERY, CLI, F (74.67) | t032(63) | t022(2); t6589(2) | | t020(1); t025(1); t910(1); t1302(1); t2892(1); t4232(1); t10016(1); t19753(1) |
| CC8-MRSA-IV | 65 (15.78) | 17 (10.18) | 48 (19.59) | 55 (84.62) | ERY, F (60.00) | t008(60) | t622(2) | | t024(1); t656(1); t19659(1) |
| CC45-MRSA-IV | 54 (13.11) | 16 (9.58) | 38 (15.51) | 0 (0.00) | All susceptible (55.56) | t026(20) | t671(13) | t015(7); t330(7) | t12321(3); t095(1); t706(1); t4264(1); t4545(1); |
| CC5-MRSA-IV | 47 (11.41) | 21 (12.57) | 26 (10.61) | 13 (27.66) | ERY, CLI (63.83) | t311(28) | t179(7) | t3235(3) | t002(1); t1215(1); t1265(1); t1282(1); t1303(1); t1345(1); t9710(1); t11862(1); t19661(1) |
| CC5-MRSA-I | 41 (9.95) | 8 (4.79) | 33 (13.47) | 0 (0.00) | ERY, CLI (63.41) | t010(33) | t045(2) | t1683(2) | t855(1); t2065(1); t19657(1); t19662(1) |
| Other clones[b] | 16 (3.88) | 4 (2.40) | 12 (4.90) | 4 (25.00) | | | | | |
| Undetermined[c] | 8 (1.94) | 0 | 8 (3.27) | 0 (0.00) | | | | | |
| Total | 412 | 167 | 245 | 72 (17.48) | | | | | |

[a] AR, antimicrobial resistance; ERY, erythromycin; CLI, clindamycin; F, fluoroquinolones (norfloxacin and/or ciprofloxacin). Detailed antimicrobial resistance profiles for all isolates can be found in the Supplemental Material.
[b] Other clones: CC398-MRSA-V (t034[3]; t4652[1]); CC398-MRSA-IV (t779[1]); CC45-MRSA-V (t065[2]; t880[1]); CC1-MRSA-IV (t127[1]; t17565[1]); CC80-MRSA-IV (t044[2], PVL +); CC88-MRSA-IV (t2393[1], PVL +; t5163[1]); CC121-MRSA-V (t314[1], PVL+); CC59-MRSA-V (t316[1]).
[c] spa types of isolates for which the SCCmec type was not determined and for which it was impossible to assign the clone: t026(1); t179(1); t311(1); t330(2); t548(1); t7603(1); t17768(1).

$P < 0.001$), but there was no significant difference between inpatients and outpatients (17 versus 47; $P = 0.087$).

Genes coding for Panton-Valentine leukocidin were found in 72 (17.48%; 72/412) isolates, besides CC8-MRSA-IV ($n = 55$), as well as in isolates belonging to CC5-MRSA-IV ($n = 13$), CC80-MRSA-IV ($n = 2$), CC88-MRSA-IV ($n = 1$), and CC121-MRSA-V ($n = 1$). These isolates frequently originated from wound swabs (50.00%; 36/72). For 65/72 (90.3%) isolates with complete data, a statistical analysis was performed (Fig. 2). In general, PVL positivity was associated with isolates causing infection (72.31%; 47/65; $P < 0.001$), outpatients (75.38%; 49/65; $P = 0.009$), and lower age (median age of 25.30 versus 54.26 years; $P < 0.001$). These isolates were typically resistant to three classes of antimicrobials (63.08%; 41/65; $P < 0.001$). The common resistance profile includes, besides antistaphylococcal beta-lactams, macrolides (87.69%; 57/65) and/or fluoroquinolones (76.92%; 50/65). However, resistance to lincosamides was much less frequent (24.62%; 16/65).

In the other frequently found lineages in our study, that is CC5-MRSA-I (9.95%; 41/412), CC5-MRSA-IV (11.41%; 47/412), and CC45-MRSA-IV (13.11%; 54/412), significant associations were found for CC5-MRSA-I and outpatients (33 of 41 isolates; $P = 0.028$). For all three of the lineages, namely, CC5-MRSA-I (median age of 20.94 years), CC5-MRSA-IV (11.16 years), and CC45-MRSA-IV (11.11 years), the age of the patient was significantly lower, compared to those observed with CC5-MRSA-II (70.78 years) and CC22-MRSA-IV (67.28 years), but not compared to that with CC8-MRSA-IV (31.47 years). These lineages were less resistant, compared to CC5-MRSA-II, CC22-MRSA-IV, and CC8-MRSA-IV. Resistance to fluoroquinolones was especially rare among these isolates. CC45-MRSA-IV was associated with resistance to antistaphylococcal beta-lactams only ($P < 0.001$).

Eight lineages were represented by four or fewer isolates (Table 2). One MRSA (t7603; CC130) isolate carrying the *mecC* gene was found.

## DISCUSSION

The Slovakia MRSA invasive infection prevalence of 24.81% (134/540) is considerably higher, compared to neighboring countries, such as the Czech Republic (9.28%; 194/2,089), Austria (4.40%; 125/2,843), and Poland (13.84%; 187/1,351) (7), but no data on the characteristics of MRSA strains are available from this region. To fill this knowledge gap, we collected 412 MRSA isolates, including inpatient and outpatient MRSA cases, covering all regions of Slovakia.

We found the most common Slovak MRSA lineages to be CC5-MRSA-II (t014, t003) and CC22-MRSA-IV (t032), and these were responsible for 25.73% (106/412) and 18.20% (75/412) of the MRSA cases in this study, respectively. These lineages belonged to the most common HA-MRSA in Central Europe (10), and they together caused 60.48% (101/167) of the inpatient cases in our study. Typical for HA-MRSA, both lineages were associated with the advanced age of patients and the resistance to multiple antimicrobials. A similar dominance of the aforementioned clones was observed in Austria, forming 36.48% (58/159) and 22.64% (36/159) among 159 MRSA isolates in 2012 (11). In 153 MRSA isolates from bloodstream infections that were collected between 2011 and 2016 in Hungary, the CC22-MRSA-IV was a dominant clone (66.67%; 102/153) and it was followed by CC5-MRSA-II (up to 23.53%; 36/153) (12). In the Czech Republic (81.41%; 359/441) and Poland (21.70%; 23/106 to 48.81%; 41/84), CC5-MRSA-II dominated (9, 13, 14). Notably, CC22-MRSA-IV was either rarely detected or not detected.

Interestingly, the third most frequently detected lineage in this study was CC8-MRSA-IV (15.78%; 65/412), with characteristic features of the US epidemic CA-MRSA clone USA300, that is, SCC*mec* subtype IVa with genes coding for PVL and *arcA* (15) in the majority of isolates (83.33%; 54/65). The onset and subsequent spread of the USA300 epidemic in America in 2005 (16) were followed by several studies in other countries that were set up to monitor the spread of CA-MRSA and/or the USA300 clone, in particular. In Europe, USA300 was found to be the most frequent (18%; 9/51) among 51 imported CA-MRSA cases that were reported between 2011 and 2016 in 7 countries (17). Other studies detecting the sporadic presence of USA 300 MRSA isolates in Europe concluded this to be associated with multiple

introductions of USA300 from abroad, with only a limited local spread (6, 18–21). Originally, USA300 was frequently found in inpatients without any obvious risk factors for MRSA acquisition, and it often caused invasive infection and exhibited limited antimicrobial resistance (4). Later, the strain acquired additional resistance genes (6) and started to be found more frequently in health care facilities (22). Correspondingly, the results of our study found that USA300 isolates were significantly more often the cause of infection, rather than mere colonization, but they were frequently resistant to erythromycin and norfloxacin/ciprofloxacin. Additionally, they were derived from both outpatient and inpatients. Importantly, the spread of this clone in the USA led to an increase in the number of MRSA infections (23), and severe forms of Staphylococcal infections, such as necrotizing fasciitis and pneumonia, became more common (4, 24). The burden of USA300 on the Slovak health care system as well as the factors driving the spread of USA300 in Slovakia are currently unknown. Further studies will be needed to assess the risks for Slovak patients and possible dissemination into neighboring countries.

PVL positivity (a marker of CA-MRSA) among all isolates was significantly associated with infection, younger age, and an outpatient setting, pointing to the higher virulence of these strains, as reported previously (4). Independently of PVL positivity, CA-MRSA characteristics, such as a younger age and an outpatient setting, were associated with CC5-MRSA-I (t010, t045), CC5-MRSA-IV (t311, t179, t3235), and CC45-MRSA-IV (t015, t026, t330, t671, t1231), but, of these, only the 11 CC5-MRSA-IV (9× t311; 2× t3235) isolates were positive for PVL. These lineages, found in between 9.95% (41/412) and 13.11% (54/412) of isolates, are rarely, if ever, found in hospital-based studies in the Czech Republic, Poland, and Austria (9, 11, 13, 14, 25–27), stressing that focusing only on hospital epidemiology could significantly underestimate the presence of strains that are circulating in the community and causing significant disease burden.

**Conclusions.** To the best of our knowledge, this is the first study on the molecular epidemiology of MRSA in Slovakia. This study confirmed the presence of the epidemic clones CC5-MRSA-II and CC22-MRSA-IV that were reported from neighboring countries and identified a high prevalence of the global epidemic CA-MRSA clone, namely, USA300 (CC8-MRSA-IV; PVL+; *arcA*+), among inpatients and outpatients. Taking into account the data from other European countries, the high prevalence of the USA300 clone in Slovakia is highly unexpected and warrants further investigation to understand the possible risk of its dissemination.

## MATERIALS AND METHODS

**Ethics.** Ethical approval and informed consent were not needed for this study because it was a laboratory-based surveillance study and because the isolates were acquired as a part of routine diagnostics. The patient data were anonymized, and the results of the study have no impact on the patient's care.

**Study design.** Between January 3rd, 2020, and March 27th, 2020, non-duplicated (single-patient) MRSA isolates were collected in Slovakia from hospitalized patients (16 hospitals) and outpatients attending outpatient hospital departments (including emergency departments), outpatient medical centers, and general practitioners (77 cities) (Fig. 1). Isolates were cultured from blood samples, samples that were collected from the site of an infection, or samples that were collected during screening for asymptomatic colonization upon the request of a physician. The MRSA screening (nose, throat, and skin swabs) was performed based on local guidelines or at the discretion of the treating physician.

The isolates were grouped as (i) invasive when isolated from a primarily sterile site (i.e., blood-culture or catheter) and/or when the isolation of MRSA from a particular site corresponded with the presence of clinical symptoms of the infection or as (ii) colonizing when the isolation of MRSA was not directly related to the patient's diagnosis.

The following anonymized data were collected: age, sex, clinical setting (inpatient or outpatient), geographic location of the health care facility, sample type, and patient diagnosis.

**Identification of the isolates and antimicrobial susceptibility testing.** The species identification of isolates was confirmed via MALDI-TOF MS Biotyper v 3.1 (Bruker Daltonics). Antimicrobial susceptibility was tested via the disc diffusion method with the breakpoints recommended by EUCAST v12.0 (28). Inducible clindamycin resistance was tested via D tests (28). Resistance to tigecycline was detected via disc diffusion and confirmed via the gradient diffusion test. The multiplicity of antimicrobial resistance was counted by using previously described antistaphylococcal drug categories (29).

Isolates were tested for the presence of the *mecA* gene and for genes coding Panton-Valentine leucocidin (PVL) via qPCR (30). In cases of *mecA* negative results, the presence of the *mecC* gene was tested via PCR (31). In addition, the CC8-MRSA-IV isolates were tested via PCR for the presence of *arcA* (15).

**Clonal analysis.** The *spa* typing was performed as previously described (32), and the individual *spa* types were assigned by using the Ridom StaphType software package, version 2.2.1 (Ridom, Germany). They were clustered into *spa* clonal complexes by using the based upon repeat pattern (BURP) algorithm. The multilocus sequence typing STs and/or clonal complexes (CC) were either inferred by using the SpaServer database (http://spaserver.ridom.de) or derived from previously published studies.

SCC*mec* types I to VI were determined by using multiplex PCR (33), and the subtyping of the IVa to IVd and IVg to IVh types was performed (34, 35).

**Statistics.** Differences between groups were evaluated by using the chi-square test for categorical variables. For small sample sizes, Fisher¨s exact test was used. A Kruskal-Wallis test and a *post hoc* Dunn's test with the Holm correction were used for the comparisons of continuous variables.

A *P* value of ≤0.05 was considered to be indicative of a statistically significant result. The analyses were conducted by using the R statistical libraries "stats" and "rstatix".

Isolates with incomplete epidemiological or demographic data as well as those originating from lineages that were represented by fewer than 10 isolates were excluded from the statistical analysis. Data for 384 out of 412 isolates were included.

## SUPPLEMENTAL MATERIAL

Supplemental material is available online only.

**SUPPLEMENTAL FILE 1**, XLSX file, 0.2 MB.

## ACKNOWLEDGMENTS

The study was supported by the project National Institute for Virology and Bacteriology (Program Exceles, ID Project No. LX22NPO5103)-Funded by the European Union-NextGenerationEU.

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
