## [Reviewer comments · Microbiology Spectrum]

Microbiology Spectrum

Epidemiology of methicillin-resistant *Staphylococcus aureus* (MRSA) in Slovakia, 2020 – Emergence of epidemic USA300 clone in community and hospitals

Jan Tkadlec, Anh Vu Le, Marie Brajerova, Anna Soltesova, Jozef Marcisin, Pavel Drevinek, and Marcela Krutova

Corresponding Author(s): Jan Tkadlec, 2nd Faculty of Medicine, Charles University and Motol University Hospital, Czech Republic

Review Timeline:

Submission Date:	March 23, 2023
Editorial Decision:	April 15, 2023
Revision Received:	May 16, 2023
Accepted:	June 1, 2023

Editor: Kunyan Zhang

Reviewer(s): Disclosure of reviewer identity is with reference to reviewer comments included in decision letter(s). The following individuals involved in review of your submission have agreed to reveal their identity: Dennis Nurjadi (Reviewer #1)

Transaction Report:

DOI: <https://doi.org/10.1128/spectrum.01264-23>

April 15, 2023

Dr. Jan Tkadlec
2nd Faculty of Medicine, Charles University and Motol University Hospital, Czech Republic
V Uvalu 84
Prague, Motol 150 06
Czech Republic

Re: Spectrum01264-23 (Epidemiology of methicillin-resistant *Staphylococcus aureus* (MRSA) in Slovakia, 2019-2020 - Emergence of epidemic USA300 clone in community and hospitals)

Dear Dr. Jan Tkadlec:

One of the reviewer additional comment:

"Given the outliers and the range of ages, the data in figure 3 as displayed by boxplot are less convincing of statistical significance."

Link Not Available

Sincerely,

Kunyan Zhang

Journals Department
Reviewer comments:

Reviewer #1 (Comments for the Author):

General comments:

The study addresses a research gap in the epidemiology of MRSA in Slovakia. Indeed, there are no epidemiological data from this region (to my knowledge). Overall, the study was well-performed and well-written. A total of 412 isolates were characterized

using PCR (for spa type, SCCmec type, PVL, arcA and mecA/C detection). Phenotypic resistance was determined using disk diffusion. The strength of the study is the coverage of the sampling area, including both the hospital (inpatient) setting and many outpatient departments. High resolution molecular typing, such as WGS would have added more depth to the analysis but considering the lack of data in the region, the presented data is valuable.

Specific comments:

- Please use line numbering, this really helps to refer to specific text location for the review process.
- Please check if the title is appropriate (2019-2020). According to the methods the isolates were collected in 2020 (see abstract - Jan to Mar 2020)
- When presenting data as percentage, please try to include the denominator. This is sometimes not included.
- Introduction first paragraph; "caused by". Maybe use "associated with" instead.
- Methods: January - March 2020 should be specified so that the study period is clear (e.g. 1st January 2020 to 30th March 2020?)
- Methods, AST: please include the version number of the EUCAST breakpoints.
- Page 4 methods: PVL=Panton Valentine leukocidin not Pantone Valetine's leukocidin, please correct.
- Methods, ACME detection; theoretically the PCR is detecting arcA, a gene, which is present in the ACME element. Consider writing arcA detection instead of ACME detection.
- Methods, any version number of the Ridom Staphytype software?
- Page 5: chi squared test instead of chi square test (please correct)
- Table 1; please indicate that the number in brackets indicate the frequency of the isolates in the table legend/footer (column spa types).
- Figure 3 is redundant; this can be moved to supplementary data.
- Figure 2: how was the multiplicity of resistances determined? Do the beta lactams count as one class? Some explanation in the figure legend would be good.

Reviewer #2 (Comments for the Author):

In this manuscript, Tkadlec et al. aimed to characterize MRSA isolates circulating in the community from outpatients and hospitalized patients in Slovakia. I hope the authors view these suggestions as being constructive and designed to help strengthen their manuscript.

Major Comments:

There are quite a few results described in the clonal structure section where the data are in the supplemental (e.g., resistance by clone). The data could be added to table 2 to show the resistance by clone, or a note might be included in the text to point the reader toward the supplemental.

Minor Comments:

Materials and Methods section: Recommend adding if isolates from emergency department count as hospitalized or outpatient.

Staff Comments:

Preparing Revision Guidelines

Please return the manuscript within 60 days; if you cannot complete the modification within this time period, please contact me. If you do not wish to modify the manuscript and prefer to submit it to another journal, please notify me of your decision immediately so that the manuscript may be formally withdrawn from consideration by Microbiology Spectrum.

Response to Reviewers

We want to thank the reviewers for the constructive criticism they provided to increase manuscript clarity and quality. We hope we were able to sufficiently respond to all reviewers' comments.

1. One of the reviewer additional comment:

"Given the outliers and the range of ages, the data in figure 3 as displayed by boxplot are less convincing of statistical significance."

Response to the reviewer: Boxplot in Figure 3 was intended to visualise the difference in age distribution among MRSA clones. We agree that boxplots may sometimes be misleading and often serve only illustrative purposes. For that, we have also embedded the value of the test statistic, which is the actual measure of significance. The value of $4.59e-24$ is in our regard highly significant, also given the smaller dataset, where the effect must be exceptionally strong to arrive at such a value.

The difference in age distribution between patients of individual clones (presented in the manuscript) was further assessed by Dunn's test with Holm correction. We agree with suggestion of the Reviewer 1 (below), that Figure 3 is redundant and the figure was moved to the Supplement.

Reviewer comments:

Reviewer #1 (Comments for the Author):

General comments:

The study addresses a research gap in the epidemiology of MRSA in Slovakia. Indeed, there are no epidemiological data from this region (to my knowledge). Overall, the study was well-performed and well-written. A total of 412 isolates were characterized using PCR (for spa type, SCCmec type, PVL, arcA and mecA/C detection). Phenotypic resistance was determined using disk diffusion. The strength of the study is the coverage of the sampling area, including both the hospital (inpatient) setting and many outpatient departments. High resolution molecular typing, such as WGS would have added more depth to the analysis but considering the lack of data in the region, the presented data is valuable.

Specific comments:

2. Please use line numbering, this really helps to refer to specific text location for the review process.

Response to the reviewer: Continuous line numbering was used throughout the manuscript.

3. Please check if the title is appropriate (2019-2020). According to the methods the isolates were collected in 2020 (see abstract - Jan to Mar 2020)

Response to the reviewer: we want to thank the reviewer for pointing out this inaccuracy, the date in the title was changed to 2020.

4. When presenting data as percentage, please try to include the denominator. This is sometimes not included.

Response to the reviewer: As suggested by Reviewer, the data presented as percentages were supplemented with the denominators throughout the article.

5. Introduction first paragraph; "caused by". Maybe use "associated with" instead.

Response to the reviewer: Changed as suggested (line 75).

6. Methods: January - March 2020 should be specified so that the study period is clear (e.g. 1st January 2020 to 30th March 2020?)

Response to the reviewer: The study period was specified as suggested (line 238).

7. Methods, AST: please include the version number of the EUCAST breakpoints.

Response to the reviewer: The version of EUCAST breakpoints was added (line 254).

8. Page 4 methods: PVL=Panton Valentine leukocidin not Panton Valetine's leukocidin, please correct.

Response to the reviewer: Corrected (line 258).

9. Methods, ACME detection; theoretically the PCR is detecting arcA, a gene, which is present in the ACME element. Consider writing arcA detection instead of ACME detection.

Response to the reviewer: As suggested we changed the ACME detection to the arcA detection throughout the article.

10. Methods, any version number of the Ridom Staphtype software?

Response to the reviewer: The version of the Ridom Staphtype software was added (line 263)

11. Page 5: chi squared test instead of chi square test (please correct)

Response to the reviewer: Corrected (line 270)

12. Table 1; please indicate that the number in brackets indicate the frequency of the isolates in the table legend/footer (column spa types).

Response to the reviewer: Corrected

13. Figure 3 is redundant; this can be moved to supplementary data.

Response to the reviewer: Figure 3 was moved to supplementary material.

14. Figure 2: how was the multiplicity of resistances determined? Do the beta lactams count as one class? Some explanation in the figure legend would be good.

Response to the reviewer: for determination of multiplicity of resistance we used the classification of antimicrobials from the definition of multidrug or pandrug-resistant phenotype of *S. aureus* from the study of Magiorakos et al 2012: Multidrug-resistant, extensively drug-resistant and pandrug-resistant bacteria: an international expert proposal for interim standard definitions for acquired resistance. In this study the antistaphylococcal beta-lactams (oxacillin and ceftioxin) count as one class.

Legend of Figure 2 was expanded to include this specification.

Legend was reworded to: Parallel coordinates describing the association of the presence of PVL and MRSA clone, level of resistance, and the origin of infection or colonisation.

Reviewer #2 (Comments for the Author):

In this manuscript, Tkadlec et al. aimed to characterize MRSA isolates circulating in the community from outpatients and in hospitalized patients in Slovakia. I hope the authors view these suggestions as being constructive and designed to help strengthen their manuscript.

Major Comments:

15. There are quite a few results described in the clonal structure section where the data are in the supplemental (e.g., resistance by clone). The data could be added to table 2 to show the resistance by clone, or a note might be included in the text to point the reader toward the supplemental.

Response to the reviewer: *The most frequent antimicrobial resistance profile with corresponding frequency was added for each clone in Table 2.*

Minor Comments:

16. Materials and Methods section: Recommend adding if isolates from emergency department count as hospitalized or outpatient.

Response to the reviewer: *Thank you for pointing to it. We changed the corresponding part to specify that the emergency departments were included among outpatient settings (line 240).*

June 1, 2023

Dr. Jan Tkadlec
2nd Faculty of Medicine, Charles University and Motol University Hospital, Czech Republic
V Uvalu 84
Prague, Motol 150 06
Czech Republic

Re: Spectrum01264-23R1 (Epidemiology of methicillin-resistant *Staphylococcus aureus* (MRSA) in Slovakia, 2020 - Emergence of epidemic USA300 clone in community and hospitals)

Dear Dr. Jan Tkadlec:

Your manuscript has been accepted, and I am forwarding it to the ASM Journals Department for publication. You will be notified when your proofs are ready to be viewed.

Sincerely,

Kunyan Zhang
Editor, Microbiology Spectrum
